# Automated interpretation of prenatal ultrasound using a predefined acquisition protocol in resource-limited countries

**Thomas L. A. van den Heuvel**[1,2]          Thomas.vandenHeuvel@radboudumc.nl

[1] *Diagnostic Image Analysis Group, Department of Radiology and Nuclear Medicine, Radboud University Medical Center, Nijmegen, the Netherlands*
[2] *Medical Ultrasound Imaging Centre, Department of Radiology and Nuclear Medicine, Radboud University Medical Center, Nijmegen, the Netherlands*

**Chris L. de Korte**[2,3]          Chris.deKorte@radboudumc.nl

[3] *Physics of Fluids Group, MIRA, University of Twente, the Netherlands*

**Bram van Ginneken**[1,4]          Bram.vanGinneken@radboudumc.nl

[4] *Fraunhofer MEVIS, Bremen, Germany*

**Editors:** Under Review for MIDL 2019

## Abstract

In this study, we combine a standardized acquisition protocol with image analysis algorithms to investigate if it is possible to automatically detect maternal risk factors without a trained sonographer. The standardized acquisition protocol can be taught to any health care worker within two hours. This protocol was acquired from 280 pregnant women at St. Luke's Catholic Hospital, Wolisso, Ethiopia. A VGG-like network was used to perform a frame classification for each frame within the acquired ultrasound data. This frame classification was used to automatically determine the number of fetuses and the fetal presentation. A U-net was trained to measure the fetal head circumference in all frames in which the VGG-like network detected a fetal head. This head circumference was used to estimate the gestational age. The results show that it possible automatically determine gestational age and determine fetal presentation and the potential to detect twin pregnancies using the standardized acquisition protocol.

**Keywords:** Prenatal, Ultrasound, Deep Learning, Resource-limited countries

## 1. Introduction

World Health Organization et al. (2014) reported that 99% of all maternal deaths occur in resource-limited countries. This corresponds to more than 800 women that die each day as a consequence of their pregnancy. Ultrasound imaging is widely used to detect maternal risk factors during the pregnancy. Unfortunately, ultrasound imaging remains often out of reach for pregnant women in resource-limited countries, since there is a lack of trained sonographers that can acquire and interpret the images. DeStigter et al. (2011) have introduced the obstetric sweep protocol (OSP). Figure 1.A shows the OSP, which consists of six predefined free-hand sweeps over the abdomen of the pregnant women. The OSP can be taught to any health care worker within two hours, which obviates the need to extensively train sonographers for months. In this work we investigated if it was possible to automatically determine the number of fetuses, estimate gestational age (GA) and determine fetal presentation using the OSP.

## 2. Methods

### 2.1. Data acquisition

The OSP was acquired from 280 pregnant women in Wolisso, Ethiopia. This data represents the target population of our research, since Ethiopia still has a high maternal mortality ratio (World Health Organization et al., 2014). A gynecologist determined the number of fetuses and measured the reference fetal head circumference (HC) in the standard plane. This reference HC was used to obtained the reference GA for each single pregnancy. The gynecologist also determined for all single pregnancies if the fetus lay in cephalic or breech presentation.

### 2.2. Automated interpretation

The automated interpretation of the OSP data consists of five steps:
First, a deep learning network, inspired on the VGG-net of Simonyan and Zisserman (2014), was trained to classify which frames within the OSP data present the fetal head, the fetal torso in the transverse direction, the fetus in sagittal direction and when the transducer was detached from the abdomen of the pregnant women in between the six sweeps (van den Heuvel et al., 2019).
Secondly, the frames in which the transducer was detached from the abdomen were used to automatically separate the six sweeps. Each sweep was resampled to 100 frames per sweep using nearest neighbor interpolation.
Thirdly, a Random Forest classifier (RFC) (Breiman, 2001) was used to determine the number of fetuses using the frame classification of the six separated sweeps.
Fourthly, a second deep learning system, inspired on the U-net of Ronneberger et al. (2015), was trained to measure the HC in all frames in which the first deep learning system has detected a fetal head completely present. Since the HC measured in the standard plane is on of the largest circumference one can measure from the fetal head, the 75th percentile of all estimated HCs was used as the final HC. The curve of Hadlock et al. (1984) was used to determine the GA from the automatically estimated HC.
Finally, a second RFC was used to determine fetal presentation using the frame classification of the six separated sweeps.
All systems were trained using a five-fold cross-validation.

## 3. Results

The data included a total of 247 single and 33 twin pregnancies for which the result of the automated system is shown in Table 1. The reference HC measurement was not acquired from 7 single pregnancies and 15 of the HCs fell outside the limits of the Hadlock curve, so a total of 225 GAs were included. The median difference between the reference GA measured in the standard plane and the automatically estimated GA measured from the OSP data was -0.4 days with an interquartile range (IQR) of 15.2 days. The median difference was -0.4 with an IQR of 9.3 days in the second trimester (14 weeks until 28 weeks of gestation). The 247 single pregnancies contained 216 fetuses in cephalic presentation and 31 fetuses in breech presentation. Table 2 shows the results of the automated detected presentation.

Figure 1.B shows an example of a fetus in breech presentation. The frame classification shows that that the fetal head (blue) lies above the fetal torso (green).

Table 1: Results for number of fetuses

| No. fetuses | Correct | Incorrect |
|---|---|---|
| Single | 244 | 3 |
| Twin | 20 | 13 |

Table 2: Results for fetal presentation

| Presentation | Correct | Incorrect |
|---|---|---|
| Cephalic | 215 | 1 |
| Breech | 31 | 0 |

## 4. Discussion

In this work we combined the OSP with automated image analysis to automatically interpret the ultrasound data, with the aim to obviate the need for a trained sonographer.

The results show that it possible automatically determine gestational age with a IQR of 9.3 days in the second trimester, which could be used to indicate in which week the health care worker should send the pregnant women to a health care facility when a maternal risk factor is detected. This is very important in resource-limited settings, since it can take more than one day for a pregnant woman to reach a hospital.

Table 2 shows that all breech presentations were correctly detected and 99.5% of the fetuses in cephalic presentation were correctly detected. When taking into account that 33 of the 247 single pregnancies showed some kind of abnormality, we conclude that the system is able to robustly determine the fetal presentation. The system could therefore give health care workers the option to refer a fetus in breech presentation to a health care facility.

Table 1 shows that 98.8% of all single pregnancies were correctly detected. The system only detected 60.6% of all twin pregnancies, because the system was not able to detect twins that lie in the same presentation (e.g. both cephalic). This shows potential to detect twin pregnancies, but it leaves room for improvement in the future.

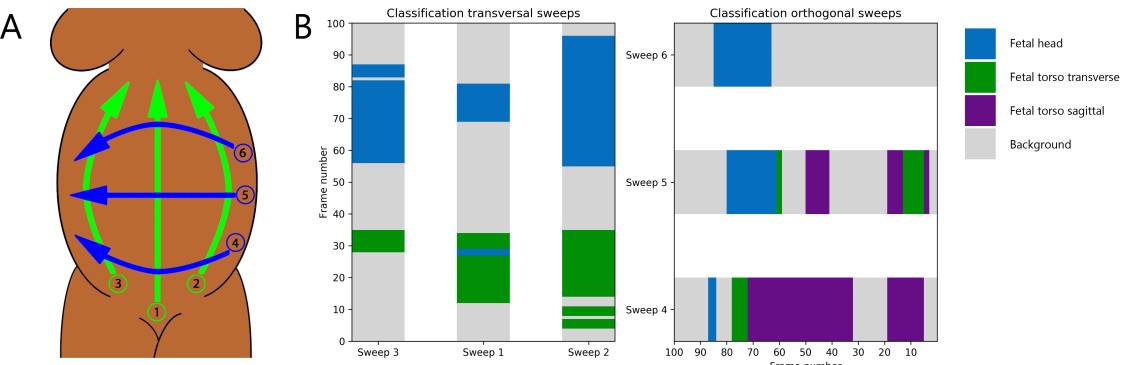

Figure 1: A. Visualization of the six sweeps of the obstetric sweep protocol.
B. Result of the frame classification, showing a fetus in breech presentation.

## Acknowledgments

We thank Dr. Hezkiel Petros and Dr. Stefano Santini from St. Luke's Catholic Hospital for their help with the acquisition of the ultrasound data that was used in this study.

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
