# OpenReview forum: "Automated interpretation of prenatal ultrasound using a predefined  acquisition protocol in resource-limited countries"
_MIDL.io/2019/Conference/Abstract — MIDL Abstract 2019_

### Official Review · AnonReviewer1 · 2019-04-29
**Important study with good initial results**

**Rating:** 3
**Confidence:** 2

**Review:**

This is an important study which evaluates the applicability of the automated methods in developing world. The experiment is well-documented and obtained good results for GA prediction and presentation classification.

I believe that for twin classification, it should be better to use image features than the classification features. Either perform single/twin classification for each frame, then do voting, or you could sample N frames per sweep and use a volume with 6N channels (e.g. 60) as an input to the network (RF or DL).

---

### Official Review · AnonReviewer2 · 2019-04-30
**Interesting approach, but apparently high overlap with previous workshop presentation**

**Rating:** 3
**Confidence:** 2

**Review:**

Authors try to automatically interpret prenatal ultrasound that has been acquired by non-experts under resource-constrained conditions. For sure, this is important and relevant to MIDL. The approach is interesting from a technical perspective, mixing different deep learning approaches and random forests in a way that appeared sensible to me. Results are also promising for detecting gestational age and fetal presentation. Authors also attempt to detect twin pregnancies, but with limited success so far.

One concern is that, in order to be practically useful, the method would have to prove an ability to transfer results between operators. I do not think that the current cross-validation based evaluation verifies this. Moreover, there are some small language errors (e.g., "The results show that it possible...").

Reviewers were told that submitting extended abstracts on recent journal papers is acceptable, but work that was previously presented at conferences is not. The authors reference a journal paper, but I also found a workshop paper by the same authors that appears to have a substantial overlap with this submission (LNCS 10549:105-112). Therefore, I would be fine with acceptance, but I would not prioritize this submission in case there should not be enough poster slots.

---

### Decision · Program_Chairs · 2019-05-06
**Acceptance Decision**

Accept